# The mechanical energetics of walking across the adult lifespan

**Bernard X. W. Liew**[1]*, **David Rugamer**[2], **Kim Duffy**[1], **Matthew Taylor**[1], **Jo Jackson**[1]

**1** School of Sport, Rehabilitation and Exercise Sciences, University of Essex, Colchester, Essex, United Kingdom, **2** Department of Statistics, Ludwig-Maximilians-Universität München, Munich, Germany

* bl19622@essex.ac.uk, liew_xwb@hotmail.com

## Abstract

### Purpose

Understanding what constitutes normal walking mechanics across the adult lifespan is crucial to the identification and intervention of early decline in walking function. Existing research has assumed a simple linear alteration in peak joint powers between young and older adults. The aim of the present study was to quantify the potential (non)linear relationship between age and the joint power waveforms of the lower limb during walking.

### Methods

This was a pooled secondary analysis of the authors' (MT, KD, JJ) and three publicly available datasets, resulting in a dataset of 278 adults between the ages of 19 to 86 years old. Three-dimensional motion capture with synchronised force plate assessment was performed during self-paced walking. Inverse dynamics were used to quantity joint power of the ankle, knee, and hip, which were time-normalized to 100 stride cycle points. Generalized Additive Models for location, scale and shape (GAMLSS) was used to model the effect of cycle points, age, walking speed, stride length, height, and their interaction on the outcome of each joint's power.

### Results

At both 1m/s and 1.5 m/s, A2 peaked at the age of 60 years old with a value of 3.09 (95% confidence interval [CI] 2.95 to 3.23) W/kg and 3.05 (95%CI 2.94 to 3.16), respectively. For H1, joint power peaked with a value of 0.40 (95%CI 0.31 to 0.49) W/kg at 1m/s, and with a value of 0.78 (95%CI 0.72 to 0.84) W/kg at 1.5m/s, at the age of 20 years old. For H3, joint power peaked with a value of 0.69 (95%CI 0.62 to 0.76) W/kg at 1m/s, and with a value of 1.38 (95%CI 1.32 to 1.44) W/kg at 1.5m/s, at the age of 70 years old.

### Conclusions

Findings from this study do not support a simple linear relationship between joint power and ageing. A more in-depth understanding of walking mechanics across the lifespan may provide more opportunities to develop early clinical diagnostic and therapeutic strategies for impaired walking function. We anticipate that the present methodology of pooling data

**Data Availability Statement:** All analyses were conducted in R software, with associated codes and results found online (https://doi.org/10.5281/zenodo.5618838).

**Funding:** The author(s) received no specific funding for this work.

**Competing interests:** The authors have declared that no competing interests exist.

across multiple studies, is a novel and useful research method to understand motor development across the lifespan.

## Introduction

Walking is a fundamental activity of daily living, the performance of which is required for independent living, and exercise [1, 2]. Significant changes occur to walking mechanics over the adult lifespan [3], that ultimately impinge on walking speed. Understanding what constitutes normal walking mechanics across the adult lifespan is crucial to the identification and intervention of early decline in walking function. Joint mechanical energetics (ME) (i.e. power and work) [4–6] are some of the most well investigated variables in walking, given that they reflect the muscular sources of energy required to walk [7, 8].

Older adults (average 69 years) have been reported to walk with 2.7 times greater hip positive work during early support, and 0.7 times less ankle positive work during push-off than young adults (average 21 years) [4]. Another study reported that older adults (average 66 years) walked with 0.9 times lesser ankle push-off power, and 1.25 times greater hip power during early support compared to young adults (average 26 years) during walking at 1 m/s [6]. Common statistical methods used to quantify age-related changes in joint ME during walking are linear regression-based techniques, which include the Analysis of Variance (ANOVA) [4, 6].

A limitation of linear regression-based techniques is the assumption that walking mechanics and performance changes linearly across the lifespan. A meta-analysis in healthy adults describe an inverted "U" shaped relationship between age and walking speed, with speed peaking in the 3rd decade of life [9]. Studies from the Baltimore Longitudinal Study of Aging reported that ankle work during walking reduced most rapidly between 30 to 60 years old [10], whilst hip work declined with multiple peaks and troughs between 60 to 92 years old [11]. Given the close relationship between joint ME and walking speed [12, 13], the evidence suggest that relationships between age and joint ME may be non-linear. To model the nonlinear relationships, Ko et al. [10] used spline regression to partition the data into three age categories, where data in each category received their own linear model fitting. A limitation of Ko et al. (2012) was that similar non-linear effects of age on joint ME was assumed for all joints (hip, knee ankle) [10].

It is well established that joint ME in walking is influenced by walking speed [13] and step/stride length [14]. Differences in joint ME with ageing could be confounded by age-related alterations in speed [9], and step/stride length [3]. To control for such confounders age-related differences in joint ME have been investigated at fixed experimental speeds [6, 15]. A limitation of experimentally controlling confounders is that it may reduce the ecological validity of the comparison. For example, fixing the walking speed of a 20-year-old to so that an 80-year-old walked at the same speed, and vice-versa, may require the former to walk in an unnatural way.

The aim of the present study was to quantify the relationship between age and joint power during walking, across the adult lifespan. To achieve this aim, we pooled together the individual participant data of three publicly available datasets [16–18], and the data from one primary research. To this end, we used a novel statistical technique called, Generalized Additive Models for Location, Scale, and Shape (GAMLSS) [19]. An advantage of GAMLSS is that it can model linear and nonlinear relationships between age and the entire joint power waveform, whilst

adjusting for potential covariates, to parse out the "true" effect of age in joint power alterations during walking. Given that the entire joint power waveform is modelled, we focus the statistical inference on three discrete parameters, namely: ankle push-off power generation (A2), hip power generation during early support (H1), and hip power generation during pre-swing (H3). These three parameters were selected due to their importance in influencing walking speed, and based on prior investigations describing their importance towards distinguishing older from younger adult gait. Between the speeds investigated presently, joint kinetics typically scale with speed—i.e. it increases in magnitudes at faster speeds [20]. Given that walking speed appears to peak at 30 years old [9], we generated the following null hypotheses. 1) A2 would exhibit an inverse "U" shaped function across age, with peaks at approximately 30 years old, similar to walking speed. Also, given that the hip and ankle power has a reciprocal relationship [4, 21], we also hypothesized that H1 and H3 would exhibit a "U" shaped function across age, with a trough happening at the age where A2 peaks.

## Methods

### Design

This was a pooled secondary analysis of the author's (MT) and three publicly available datasets [16–18]. Hence, no ethical approval was required for the conductance of this secondary analysis. Despite the presence of some methodological variations between the presently included studies, data pooling was deemed appropriate to conduct based on several reasons. First, a previous meta-analysis [3] pooled data into a random-effects model despite methodological variations in the primary studies (e.g. barefoot walking [6] and shod walking [22]). The present analysis also adopted a random effects modelling approach. Second, a previous study reported no significant differences in A2 and H3 powers between treadmill and overground walking [23]. Third, given that walking without shoes reduces step length compared to shoes [24], to account for between-study variation in footwear presently, we included step length as a covariate in our models. An overview of the methodologies of the included studies can be found in S1 Table.

**Study (KD, JJ, MT—Termed simply as "taylor").**   All participants were recruited from local communities. The inclusion criteria for this study were as follows; all participants had to live independently, be independent walkers (able to walk at least 10 m unaided), with no surgical procedures occurring in the last six months, and aged fifty-five years of age or older. 140 community-dwelling older adults volunteered for the study. Ethical approval was granted by the University Ethics Committee, and all participants provided written informed consent prior to study enrolment.

Participants performed shod overground walking across a 10m walkway at their self-selected spontaneous walking speed, over a single in-ground force plate (Kistler 9281CA, Winterthurm, Switzerland). Five successful trials (i.e. no observable targeting of force plate) were captured for each participant. A 7-camera VICON T20 motion capture system (Vicon, Oxford, UK, 100 Hz) synchronised to a force plate (1000Hz) was used. Sixteen reflective markers were placed on the lower body (7 segments, each 3DOF) body in accordance with the Plug-In Gait (PiG) marker set [25]. Processing was performed using Vicon Nexus (v 1.8.5, Oxford, UK).

Marker trajectories were filtered with a quintic spline filter (Woltring; mean square error of 10) [26], whilst force data were filtered using a low-pass 10Hz order Butterworth filter. A force plate threshold of 10N was used to determine gait events of initial contact and toe-off. Walking speed was measured via Brower Timing gates (Utah, USA) positioned 2.3 m apart, either side of the force plate.

**Study (Fukuchi) [16].** These data came from a public dataset of 42 healthy adults walking on a treadmill, the details of which can be found in the original open source publication [16]. Nine out of the 42 participants from the walking dataset were excluded from the present study. These participants had simultaneous bilateral foot contacts on the same force plate, resulting in an absence of consecutive good foot contact strides which lasted >50% of the walking duration. The 50% threshold was determined by the authors to minimize manual identification of foot contact events, to increase processing replicability [27].

Participants performed unshod walking on a dual-belt, force-instrumented treadmill (300 Hz, FIT; Bertec, Columbus, OH, USA), and motion was captured with 12 opto-electronic cameras (150Hz, Raptor-4; Motion Analysis Corporation, Santa Rosa, CA, USA) [16]. Walking occurred over eight controlled speeds: 40%, 55%, 70%, 85%, 100%, 115%, 130%, 145% of each participant's self-determined dimensionless speed (Froude number). The associated absolute walking speeds for all eight conditions for each participant were reported by the authors. Only data from the speed condition of "100%" were extracted from the present analysis. Marker trajectories and ground reaction force (GRF) were low passed filtered at a matched frequency of 6Hz (4$^{th}$ Order, zero-lag, Butterworth) [27]. A seven segment lower limb, 6DOF joint model was developed in Visual 3D software (C-motion Inc., Germantown, MD, USA) [27]. A force plate threshold of 50N was used to determine gait events of initial contact and toe-off.

**Study (Horst) [17].** These data came from a public dataset on overground walking, the details of which can be found in the original open source publication [17]. Fifty-seven participants performed unshod walking on a 10-m level walkway across two in-ground force plates (1000 Hz, Kistler, Switzerland), and motion was captured with 10 opto-electronic cameras (250 Hz, OQUS 310, Qualisys, Sweden). Participants were instructed to walk at their self-selected spontaneous speed. Walking speed was extracted by the mean anterior velocity of the modelled centre of mass (COM) during the periods when the participant was walking over the force plates. Marker trajectories and GRF data were low passed filtered (4$^{th}$ Order, zero-lag, Butterworth), at 6Hz and 18Hz, respectively. A 13-segment full body, 6DOF joint model was developed in Visual 3D software (C-motion Inc., Germantown, MD, USA) [17]. A force plate threshold of 20N was used to determine gait events of initial contact and toe-off.

**Study (Schreiber) [18].** These data came from a public dataset on overground walking, the details of which can be found in the original open source publication [18]. Fifty participants performed unshod walking on a 10-m level walkway across two in-ground force plates (1500 Hz, OR6-5, AMTI, USA), and motion was captured with 10 opto-electronic cameras (100 Hz, OQUS4, Qualisys, Sweden). Participants were instructed to walk at five speeds: 0–0.4 m/s1, 0.4–0.8 m/s, 0.8–1.2 m/s, self-selected spontaneous, and fast speeds. Only data from the self-selected spontaneous speed condition were extracted from the present analysis. Walking speed was extracted by the mean anterior velocity of the modelled COM during the periods when the participant was walking over the force plates. Marker trajectories and GRF data were low passed filtered (4$^{th}$ Order, zero-lag, Butterworth), at 6Hz and 18Hz, respectively. A 12-segment full body, 6DOF joint model was developed in Visual 3D software (C-motion Inc., Germantown, MD, USA). A force plate threshold of 20N was used to determine gait events of initial contact and toe-off. Two participants were excluded after exploratory plots of the raw power waveforms revealed larger outlier values relative to the participants across all four studies.

**Common processing across four studies.** Scalar joint power was calculated by the dot product of joint moment and angular velocity. Joint power from each joint and limb was time normalised to 101 cycle points, between two consecutive initial contacts of each limb; and was subsequently normalised to body mass (kg). For each participant and speed, the average joint power across multiple strides was calculated.

## Approach to statistical analyses

All analyses were conducted in R software, with associated codes and results found online (https://doi.org/10.5281/zenodo.5618838).

**Overview.** In statistics, classical mean regression is used to model the expectation of an outcome variable through linear effects of the covariates (also termed in the literature as predictors/independent variables). This class of models is known as Generalized Linear Models (GLMs; [28]). Generalized Additive Models (GAMs) are a more flexible class of models which allows the outcome to be modelled against the non-linear effects of the covariates [29]—a more realistic model assumption in practice. While GAMs explicitly model the expectation of the mean of the outcome, it generally does not consider the dependency of other distribution characteristics of the outcome (e.g. variance) on the available covariate information. The outcome might, for example, exhibit a larger variance (i.e. heteroscedasticity) for different observed values in time and an appropriate model must account for this by also specifying a relationship between the time information and the scale (e.g. variance) of the distribution (and not only the mean). An extension of GAMs that also accounts for uncertainty about the scale of distribution of the outcome are GAMs for location, scale and shape (GAMLSS) [19].

**Model definition.** For each of the three joints we use a GAMLSS model, where we model the mean ($\mu$) and variance ($\sigma$), of the distribution using an additive predictor:

$$y_{ij} \sim D\Big(\mu = \eta_{\mu,ij},\ \sigma = \exp\Big(\eta_{\sigma,ij}\Big)\Big),$$

where $y_{ij}$ is the power value of the respective joint of the $i$th subject and $j$th gait cycle point, $D$ the chosen distribution and $\eta_{\mu,ij}$, $\eta_{\sigma,ij}$ the respective linear predictors for the two distribution parameters. For the mean of each joint power's distribution, we choose the distribution $D$ based on the predictive performance and the linear predictor $\eta_\mu$ using Bayesian optimization (BO; [30], see details below). For the variance of each joint power's distribution, exploratory plots suggest that the cycle covariate (i.e. time normalized points) is mainly responsible for the heterogeneity in residuals when fitting GAMs with a constant variance assumption. We thus use a smooth effect of cycle for the linear predictor, i.e., $\eta_{\sigma,ij} = \beta_\sigma + f_\sigma(\text{cycle}_{ij})$, with intercept $\beta_\sigma$ and cubic regression spline $f_\sigma(\text{cycle}_{ij})$ for all joints.

**Bayesian optimization.** In the present study, the outcome represented power, whilst the following variables were used as covariates:—sex (male or female), age (years), speed (m/s), height (m), stride length (m). For the knee and hip joints, the entire stride cycle (101 data points) was included as a covariate. For the ankle, we included only a subset of data points (points 21 to 69) of the stride cycle, as a covariate. The inclusion of a reduced subset of the cycle covariate for the ankle was due to the ankle power values between 0–20% and 70–100% of the stride cycle being close to zero, which causes the model's estimates to be biased towards zero.

In order to find a suitable distribution and linear covariates for the distribution's mean, we defined a complex covariate with univariate smooths, bi-variate and tri-variate tensor product smooth for each joint (details below). Apart from the smooth effects, we included sex as dummy-encoded linear effect, a study-specific random effect, and a subject-specific random effect into the models. The study-specific random effect accounts for study specific differences in experimental protocols and processing. We then use Bayesian Optimization (BO) for covariate selection by searching through all possible combinations of basis dimensions for each of the smooth effects using a scaled t-distribution working assumption. We also allow smoothing terms to be removed completely from the linear covariate. We excluded bivariate effects only if one of the corresponding univariate effects is removed by the BO and exclude the tri-variate

effect if one of the corresponding bivariate or univariate effects is not in the linear covariate. The data were partitioned into a model training set (two-thirds of sample size) for model training, and into a testing set (one-third of sample) for performance evaluation for the BO. As performance criterion, we choose the relative root mean squared error (relRMSE) between the observed and predicted power waveform [31]. After choosing the relRMSE-optimal covariates, we compared 29 available location, scale, shape (LSS) distributions using the BO-optimal $\eta_{\mu,ij}$ and $\eta_{\sigma,ij}$ as defined in the previous section, again on the basis of the relRMSE (see S2 Table for distributions used).

For BO we choose a maximum of 300 samples from the objective function, which were then used to find an optimal setting using Gaussian Processes. Computations were performed on 2 Servers with 64GB RAM and 16 Intel(R) Xeon(R) CPU E5-2650 v2 @ 2.60GHz cores each and took up to 3 days.

**Inference.** The flexibility of GAMLSS modelling comes at a cost of losing the simplicity of reporting P-values—like in ANOVAs. The recommended approach for statistical inference is visualizing the 95% confidence interval (CI) of the partial effect of each smooth on the outcome, and predicted mean at given values of each covariate. In the present study, we reported the partial univariate effect of age with 95%CI on joint power. In addition, we also report the predicted mean joint power waveform given the following values of the varying covariates: two speeds (1, 1.5 m/s), age (20, 30, 40, 50, 60, 70, 80 years), and at a fixed stride length (1.5 m). The speeds (1, 1.5 m/s) and stride length used for prediction were selected to match the speeds and stride length observed in previous studies [4, 6, 22], to facilitate comparisons. Lastly, as specified in our hypotheses, we reported the mean and 95%CI of the age-associated peak values of A2, H1, and H3.

## Results

A total of 278 participants from all four studies were included in the present analysis. Basic descriptive summaries of the cohort can be found in Table 1 and in Fig 1. The optimal basis dimensions for each of the smooth effects, and the selected smoothing effects from the BO is reported in Table 2. For the comparison of distributions, the normal distribution turned out to be the relRMSE-optimal choice for each joint. The average relRMSE and correlation between the fitted and observed powers were 0.11 and 0.94 for the ankle, 0.13 and 0.79 for the knee, and 0.17 and 0.80 for the hip.

Fig 2 depicts the modelled smooth effect of age against joint power, which can be interpreted as the main effect of age on the average power marginalized across the gait cycle. The clearest trend with age was the knee, which saw a shift from an average positive power at 19 years old, to an average negative value peaking at -0.09 (95%CI -0.12 to -0.06) which occurred at 68 years of age, followed by a shift back to an average positive power thereafter. The smooth effect of age on power had no clear trends for the ankle and hips joints, where the 95%CI included the zero value across the age spectrum investigated.

The predicted mean joint power waveforms can be found in Fig 3. At both 1m/s and 1.5 m/s, A2 peaked at the age of 60 years old with a value of 3.09 (95%CI 2.95 to 3.23) W/kg and 3.05 (95%CI 2.94 to 3.16), respectively (Fig 4). For H1, joint power peaked with a value of 0.40 (95%CI 0.31 to 0.49) W/kg at 1m/s, and with a value of 0.78 (95%CI 0.72 to 0.84) W/kg at 1.5m/s, at the age of 20 years old (Fig 4). For H3, joint power peaked with a value of 0.69 (95% CI 0.62 to 0.76) W/kg at 1m/s, and with a value of 1.38 (95%CI 1.32 to 1.44) W/kg at 1.5m/s, at the age of 70 years old (Fig 4).

**Table 1. Descriptive characteristics (mean [standard deviation] for continuous variables) of participants.**

| Variables | fukuchi (N = 33) | horst (N = 57) | schreiber (N = 48) | taylor (N = 140) |
|---|---|---|---|---|
| Age (yo) | 39.42 (17.87) | 23.12 (2.73) | 38.17 (13.97) | 65.40 (6.47) |
| Age (min) | 21 | 19 | 19 | 55 |
| Age (max) | 84 | 30 | 67 | 86 |
| Height (m) | 1.67 (0.12) | 1.74 (0.10) | 1.74 (0.09) | 1.68 (0.09) |
| Mass (kg) | 67.66 (12.44) | 67.93 (11.26) | 71.96 (12.19) | 74.03 (14.92) |
| Sex-F | 15 (45%) | 29 (51%) | 23 (48%) | 90 (64%) |
| Sex-M | 18 (55%) | 28 (49%) | 25 (52%) | 50 (36%) |
| Speed (m/s) | 1.23 (0.17) | 1.45 (0.10) | 1.16 (0.14) | 1.41 (0.19) |
| Stride length (m) | 1.22 (0.14) | 1.51 (0.06) | 1.28 (0.12) | 1.47 (0.16) |

## Discussion

An early decline in normal walking function may result in an undesirable early loss of social independence [32]. Understanding normal age-related alterations in walking mechanics across the lifespan, is fundamental towards the development of early diagnostic and therapeutic strategies for impaired walking function. We hypothesized that A2 would exhibit an inverse "U" shaped function across age; and that H1 and H3 would exhibit a "U" shaped function across age. Our hypotheses were partially supported with H1 demonstrating a "U" shaped function

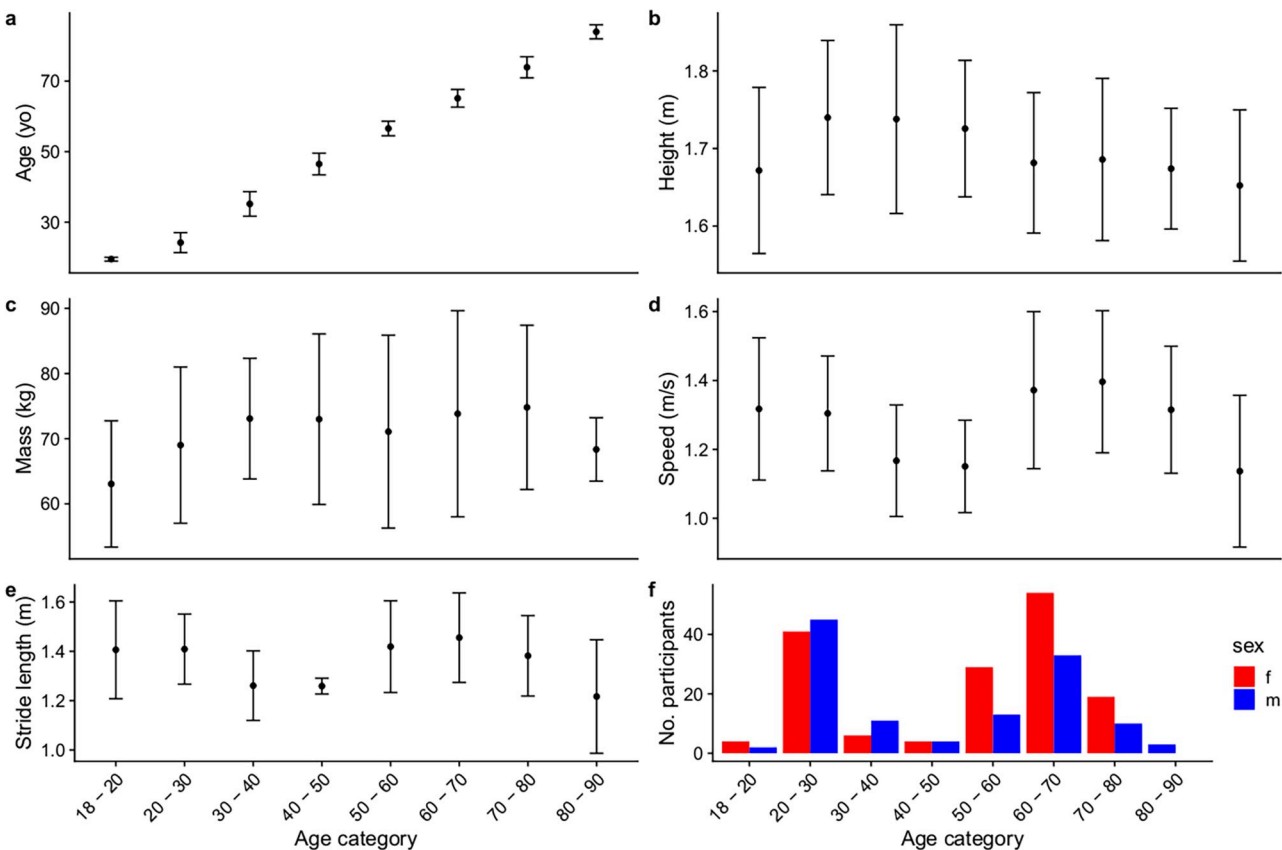

**Fig 1. Descriptive characteristics by age stratum.** a–e reflects the mean (standard deviation) of the variables for individuals within the age bracket; f reflects the number of participants by sex in each age bracket.

**Table 2. Predictors and associated basis dimension, k, of the smooths for each variable selected by Bayesian optimization.**

|  | Ankle | Knee | Hip |
|---|---|---|---|
| **f(cycle)** | 23 | 14 | 24 |
| **f(age)** | 18 | 7 | 18 |
| **f(speed)** | 9 | 20 | 10 |
| **f(ht)** | 12 | 13 | 10 |
| **f(strlen)** | 14 | 14 | 15 |
| f(cycle, age) | 12, 9 | 14, 6 | 14, 11 |
| f(cycle, speed) | 10, 4 | NA | 21, 6 |
| f(age, speed) | 5, 3 | NA | NA |
| f(cycle, ht) | 19, 4 | 16, 8 | 20, 4 |
| f(cycle, strlen) | 12, 12 | 11, 11 | 14, 14 |
| **f(cycle, age, speed)** | NA | NA | NA |

Entries with "f()" correspond to smooth effects, the subject-specific smooth of cycle is denoted with an index "subject". NA values indicate that the corresponding term was removed from the predictor. Values separated with comma indicate the different dimensions for bi- or tri-variate smooths with order corresponding to the respective term listed in the left column.

across age, but the minimum value did not occur in the younger age group. In contrast, H3 had a minimum value at the age of 30 years old in support of our hypothesis. Lastly, A2 did not demonstrate a clear maxima at a younger age.

The most surprising finding of the present study was that A2 peaked at 60 years old, which challenges the conventional thinking of a simple age-related linear decline in ankle push-off power [4, 6, 33, 34]. Differences between studies in the age-A2 relationship may be primarily attributed to the differences in statistical approaches. Existing studies that investigated the age-related decline in joint powers performed statistical inference on discrete values (e.g. A2) [4, 6, 35]. A limitation of discrete value inference techniques is that it does not consider that a signal at one cycle point can be influenced by the same signal at a prior cycle point [36]. Accounting for the time-dependency of biologic signals may be particularly important for ankle power given that a significant proportion of A2 power arises from the stretch-shortening behaviour of the Achilles tendon [37]. Given that Achilles tendon stiffness has been reported to decline with age [38], the age-related reductions in A2 could be confounded by a lower A1 power absorption in the older than younger participants [6]. The present technique, GAMLSS, was able to account for the time-dependency of the joint power waveforms by adjusting for the confounding effect of different time points.

Qiao and Jindrich proposed that joints/muscle groups could take on four different mechanical functional roles—spring, motor, damper, and strut [39]. It may be that a within-cycle adjusted A2 power has a different mechanical functional representation from the unadjusted raw A2. A joint's total positive power could be derived from recycling energy from the elastic components of the muscle-tendon unit (i.e. joint as a spring), and/or purely from the concentric activity of a muscle (i.e. joint as a motor) [39]. Speculatively, the raw A2 variable may represent the ankle's total power. Also, the within-cycle adjusted A2 may present power derived from the motor-function of the joint, since what is left after adjusting away for negative power, is the main effect of peak positive power. As previously mentioned, given the decline in Achilles tendon stiffness with age [38], our results could be interpreted as an augmentation of ankle motor-function with age for propulsion. If aging results in an elevation of ankle motor-

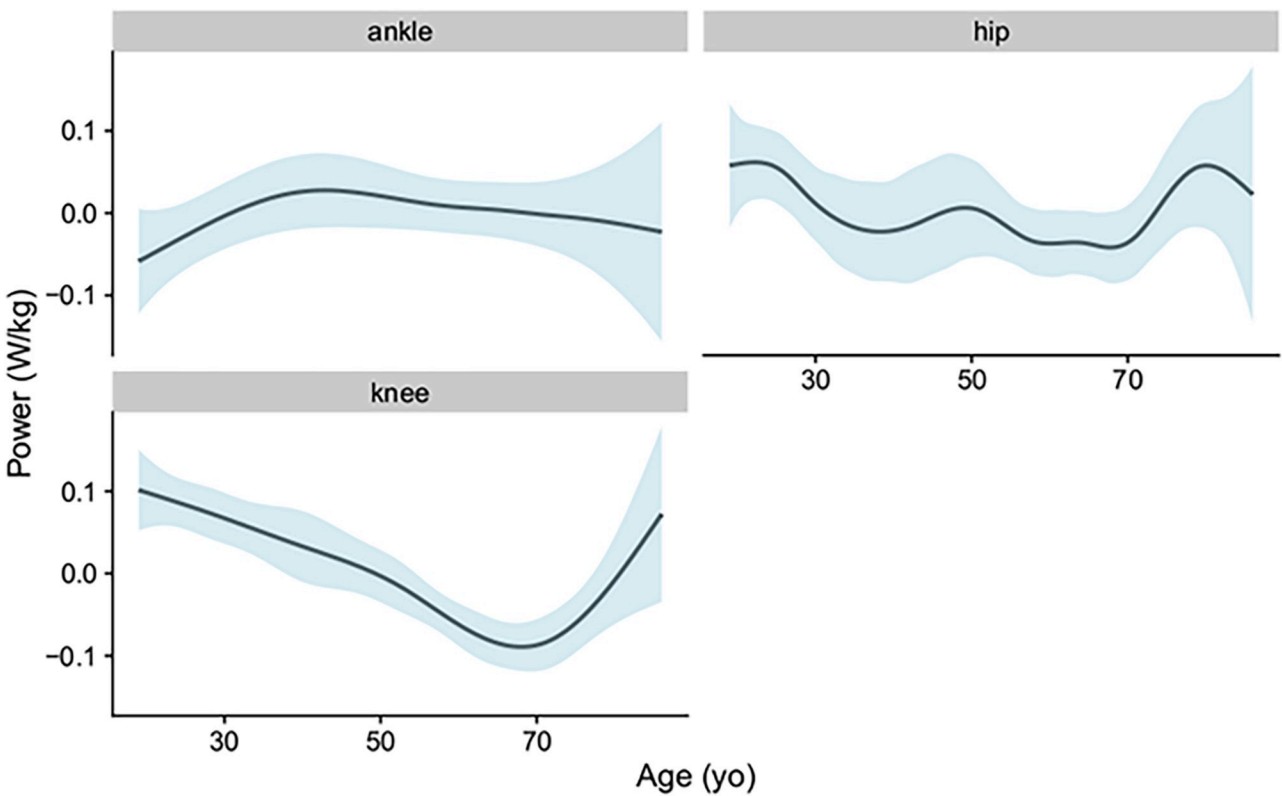

**Fig 2. Partial smooth effect of age on joint power for each joint with 95% confidence intervals of effects as shaded areas.**

function, at the expense of spring-function, this could explain the decline in mechanical efficiency in walking with increasing age [40].

The knee has not been thought of as an active energy source for propulsion in walking [41], but is important for shock absorption, joint stability, and inter-segmental energy transfer. The shift in knee power from an average positive from 19 years old to an average negative value peaking at 68 years old (Fig 2), potentially reflects an age-related biasing of muscle absorption over muscle generation. Greater negative than positive work with aging could suggest that the knee is behaving more like a damper with age [39], with the ensuring result that more positive work has to be performed by adjacent muscles to maintain walking speed. After 70 years old, the shift in knee average power from negative to positive coincides with the decline in A2 power (Fig 4), suggesting that the knee may be compensating for age-related propulsive deficits from the ankle. Evidently, research into how age influences the different mechanical functions within the lower-limb joints may provide a better understanding of what impairments drive a decline in walking performance.

In addition to adjusting for potential confounding of different time points within a waveform signal, different joint power-age relationships from the literature could arise from the presence of statistical adjustment of covariates. The present study statistically adjusted for the covariates of sex, stride length, speed, and height during all analyses. One study reported that a significantly greater A2 observed in younger than older adults was removed after adjusting for step length [14]. Another study also reported that ankle positive work in stance was greater in younger than older adults but was removed after adjusting for leg strength [42]. It may be

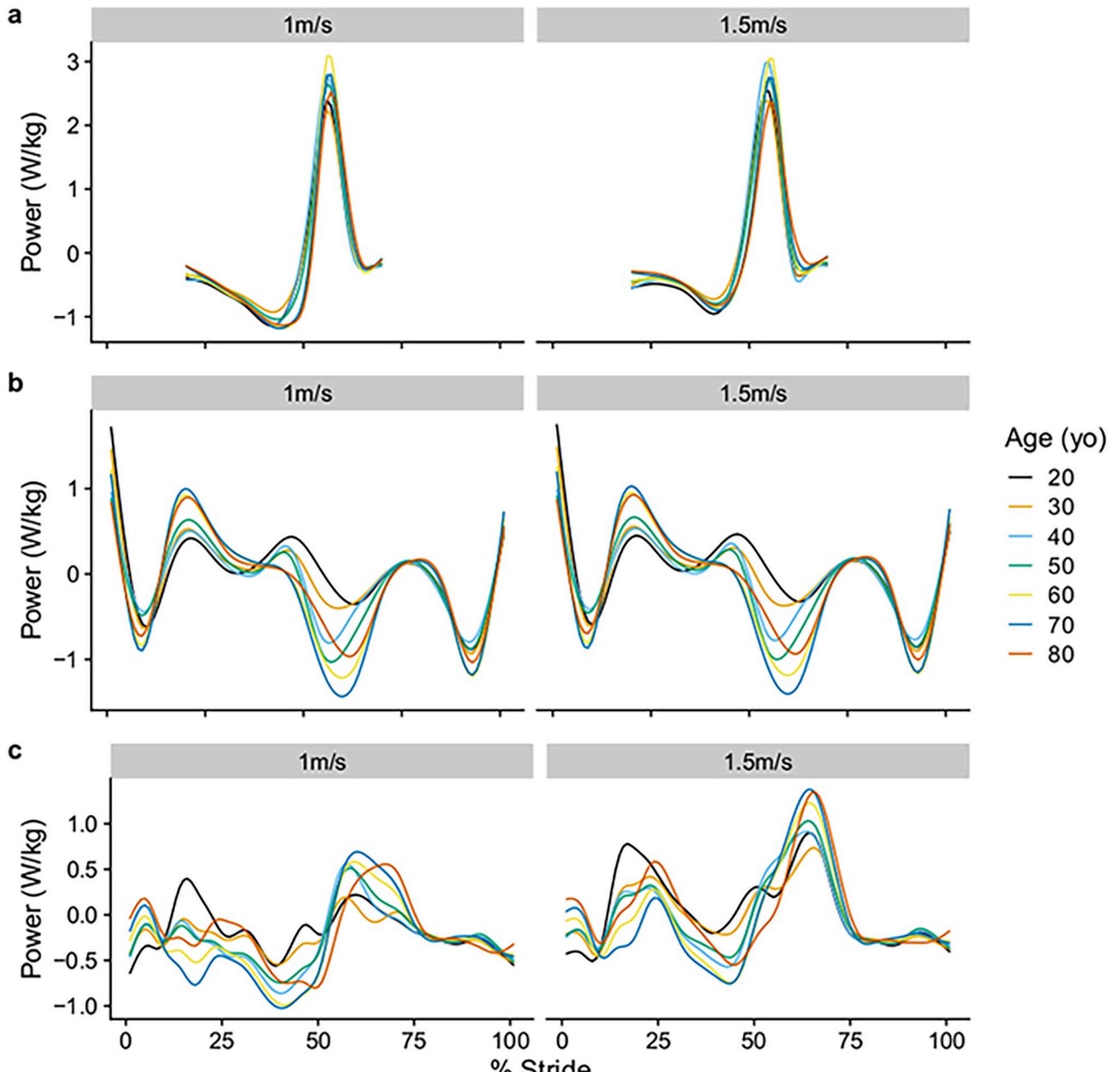

**Fig 3. Predicted mean joint power waveform from the GAMLSS model at two walking speeds (1 and 1.5m/s), at each of the seven age groups, at a fixed stride length of 1.5m.** (a) Ankle power, (b) Knee power, (c) Hip power.

argued that appropriate consideration of confounders is required in future research to accurately parse out the true relationship between joint power and age.

Differences between the present study and that of the wider literature on the age-A2 relationship could be attributed to the differences in the physical capacity of the population investigated. In the present cohort, individuals between 20–30 years walked at a self-selected mean speed of 1.30 m/s, whilst those between 50–60 years walked at a self-selected mean speed of 1.37 m/s. These speeds were comparable to Cofre et al. [6] (self-selected speed of 1.38 m/s for younger [mean 26.6 years] and 1.39 m/s for older [mean 66.8 years] groups); slower than

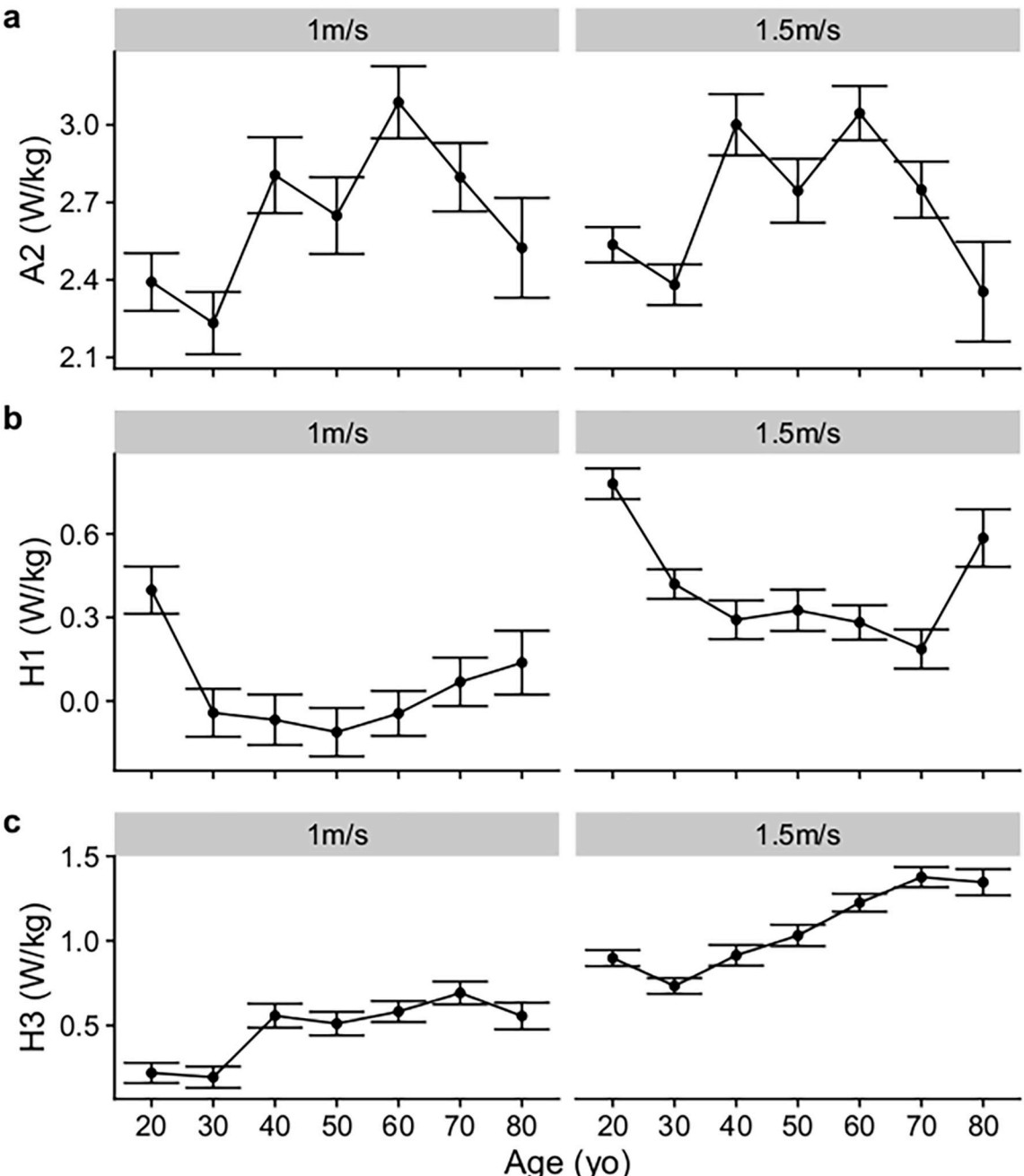

**Fig 4. Predicted mean and 95% confidence interval of A2 ankle push-off power, H1 hip power generation at early support, and H3 power generation at pre-swing from the GAMLSS model at two walking speeds (1 and 1.5m/s), at each of the seven age groups, at a fixed stride length of 1.5m.**

Kulmala et al. [22] (1.6m/s for all three age groups ["young" mean 26 years; "middle-aged" mean 61 years, and "old" 78 years], and partially faster than McGibbon and Krebs [35] (1.32 m/s for individuals "< 50" [mean 29.7 years] and 1.16 m/s for those "> 50" [mean 71.1 years]). Interestingly, Kulmala et al. [22] reported that A2 power declined only in their old cohort compared to both their young and middle-aged cohorts, without significant difference between

their young and middle-aged cohort—a finding partially consistent with the present findings. Surprisingly, individuals in their 3$^{rd}$ and 4$^{th}$ decade of life in the present study walked at a slower speed (mean 1.17, and 1.15 m/s) than those in their 2$^{nd}$ and 5$^{th}$ decade (Fig 1). Data of individuals from the 3$^{rd}$ and 4$^{th}$ decade of life came largely from two studies [16, 18], and the slower speeds in these cohorts could have influenced the present findings of a peak A2 power at 60 years old.

It has been commonly thought that in compensation for reduced A2 power, hip power generation is greater in older than younger adults [4]. However, no studies to date have reported the age-related trajectory of joint powers across the adult lifespan. Presently, we observed a "U" shaped function between age and H1, where H1 decreased between 20 to 50 years old at 1 m/s walking and increased in magnitude thereafter (Fig 4). At a faster walking speed of 1.5 m/s, the H1 continued to decline to reach a minimum at 70 years old (Fig 4). Our findings were in partial contrast to previous studies where one study found that H1 was greater in older than younger adults across speeds (1.0 to 1.6 m/s) [6]; whilst two others reported no age-related differences [22, 35]. In the present study, we found that the "U" shaped function observed for H1 was also observed with H3. The lowest magnitude of H3 was observed at 30 years old (Fig 4). H3 then peaked and plateaued at 40 years old at a speed of 1 m/s but continued increasing till 70 years old at a faster speed of 1.5 m/s (Fig 4). One study reported greater H3 in older than younger adults at faster speeds (> 1.4 m/s), but the age-related differences in H3 were not replicated by another where participants walked with a similar speed of 1.6 m/s [22].

The present study has several limitations. The secondary analysis nature of the present study means that the conclusions of our analysis are only as robust as the studies included. From Fig 1, it can be observed that individuals between 30 to 50 years old and 70 years and beyond were relatively underrepresented. The underrepresentation of individuals in these age groups explains the larger confidence intervals in mean value estimates, compared to age groups with greater sample sizes. Future research collaboration opportunities which augment data in the underrepresented age groups would be useful to provide an updated revised estimate of the relationship between age and joint power. A second limitation of the present analysis was the inclusion of studies with heterogeneous experimental set-ups, such as overground [18] versus treadmill walking [16], shod (taylor data) versus unshod conditions [18]. However, between-study heterogeneity in experimentation and indeed between-subject heterogeneity is a common occurrence in secondary analysis studies, such as in a meta-analysis [3]. We mitigated this issue by including study-specific and subject-specific random effects into our models. This meant that our predictive estimates were marginalized over different experimental study protocols and participants.

Much research has been undertaken to develop strategies to augment A2 power to optimize walking speed in older adults. Strategies such as ankle exoskeletons [43], muscle power training [44, 45], and even gait biofeedback training [46] have been developed to either overcome or augment a deficient A2 power. Based on the present study, healthy adults may not require additional therapeutic interventions to replace/augment A2 given that it is not deficient per se. We are keenly aware that this is antagonistic to current clinical recommendations, and that given the limitations on potential underrepresentation of participants in some age groups, our findings need to be replicated and augmented by future research. Although the present study did not quantify elastic energy recovery and metabolic cost, speculatively a greater A2 power in older than younger adults may reflect a less efficient movement strategy to walk at an identical speed [47]. In addition, a greater A2 power could contribute to greater dynamic postural instability in older than younger adults [48], which could explain why healthy older adults walk at a slower speed than younger adults [49].

## Conclusions

Findings from this study do not support a simple linear relationship between joint power and ageing, after adjusting for the covariates of cycle points, speed, stride length, and height. In contrast to most studies, ankle push-off power peaked at 60 years old when walking at speeds between 1 to 1.5 m/s. In addition, hip power generation at early-stance (H1) peaked at 20 years old, whilst hip power generation at pre-swing (H3) peaked at 70 years old. We adopted a novel statistical technique to model the lifespan alterations of joint power waveforms on four data-sets—the largest and most in-depth analysis to date. A more in-depth understanding of walking mechanics across the lifespan may provide more opportunities to develop early clinical diagnostic and therapeutic strategies for impaired walking function.

## Supporting information

**S1 Fig. Raw individual joint power waveforms for each participant at each walking speed.**
(PDF)

**S2 Fig. Mean joint power waveforms for each age category at each of the three speed categories.**
(PDF)

**S1 Table. Brief methodologies of included studies.**
(DOCX)

**S2 Table. 29 location, scale, shape, distributions used in Bayesian optimization.**
(DOCX)

## Author Contributions

**Conceptualization:** Bernard X. W. Liew.

**Data curation:** Bernard X. W. Liew, Kim Duffy, Matthew Taylor.

**Formal analysis:** Bernard X. W. Liew, David Rugamer, Kim Duffy.

**Funding acquisition:** Matthew Taylor, Jo Jackson.

**Investigation:** Bernard X. W. Liew, Kim Duffy, Matthew Taylor, Jo Jackson.

**Methodology:** Bernard X. W. Liew, David Rugamer, Matthew Taylor.

**Project administration:** Bernard X. W. Liew, Kim Duffy, Matthew Taylor, Jo Jackson.

**Resources:** Matthew Taylor, Jo Jackson.

**Software:** Bernard X. W. Liew, David Rugamer.

**Supervision:** Matthew Taylor, Jo Jackson.

**Validation:** Bernard X. W. Liew, Kim Duffy.

**Visualization:** Bernard X. W. Liew, David Rugamer.

**Writing – original draft:** Bernard X. W. Liew, David Rugamer, Kim Duffy, Matthew Taylor, Jo Jackson.

**Writing – review & editing:** Bernard X. W. Liew, David Rugamer, Kim Duffy, Matthew Taylor, Jo Jackson.

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
