## [Decision Letter · Decision Letter 0]

18 Aug 2021

PONE-D-21-18361

The mechanical energetics of walking across the adult lifespan.

PLOS ONE

Dear Dr. Liew,

Thank you for submitting your manuscript to PLOS ONE. After careful consideration, we feel that it has merit but does not fully meet PLOS ONE’s publication criteria as it currently stands. Therefore, we invite you to submit a revised version of the manuscript that addresses the points raised during the review process.

ACADEMIC EDITOR: Please follow the reviewers' suggestions especially with paying your attentions on clarification/justification on methodologies.

We look forward to receiving your revised manuscript.

Kind regards,

Kei Masani

Academic Editor

PLOS ONE

Journal Requirements:

2. Please include information in your Data availability statement on how all datasets used can be accessed by other researchers, particularly the author's own existing dataset

Reviewers' comments:

Reviewer's Responses to Questions

**Comments to the Author**

1. Is the manuscript technically sound, and do the data support the conclusions?

Reviewer #1: Partly

Reviewer #2: Partly

2. Has the statistical analysis been performed appropriately and rigorously? 

Reviewer #1: N/A

Reviewer #2: Yes

3. Have the authors made all data underlying the findings in their manuscript fully available?

Reviewer #1: Yes

Reviewer #2: No

4. Is the manuscript presented in an intelligible fashion and written in standard English?

Reviewer #1: Yes

Reviewer #2: Yes

5. Review Comments to the Author

Reviewer #1: I think this is a great effort to elucidate the (linear/non-linear) relationship between ageing and gait mechanics "shift" as a continuum. The secondary data analysis approach is novel and worthy considering that several studies have already collected data to test similar hypotheses. Congratulations.

However, although considered and mitigated (not mentioned to what extent) I have concerns in regards to the mixture of methodologies. There is 1 shod/treadmill study, 1 unshod/overground and 1 shod/overground; both shoes and treadmill have an effect on walking mechanics. I think a stronger argument to put these studies together is needed. To my knowledge, there is more evidence against than for putting studies with different methodologies together. Also I think that a table with full description of participants and methodologies, for comparison purposes, should be provided. What data set provided more information for the youngest and the oldest group? averaged power curves are quite different particularly at the hip (also the knee).

I think that the authors hypothesis of a inverted U shape for joint power/work in relation to aging should be better explained considering that gait maturation occurs before (potentially the ascending part of the curve) the age range included.

I partly disagree with the term "unnatural" (line 74) when asking subjects to walk at specific speeds considering that data collected in the lab is usually collected for short periods (5-8 strides unless on a treadmill), which may be similar to speed changes (range) in daily life when dealing with different street or home contexts.

Reviewer #2: This study investigated the effect of aging on the lower limb joint power using (non) linear statistical technique GAMLSS. The secondary analysis using GAMLSS method indicated that A2 peaks showed peak at 60yrs, which has not been supported by previous studies. I think the authors should discuss this result more from the biomechanical aspects. Other comments are as follows.

Introduction

１）What was the rationale behind hypothesizing inverse U shaped function for A2 peak and U shaped function for H1 and H3 peaks across age? Supporting literature is needed for these hypotheses.

Methods

2) Table 1: Please add the range of the age of participants in each study.

3) Common processing across four studies: Each participant would conducted multiple walking trials. Was the joint power data averaged across the walking trials?

4) Model definition: what does “j” mean?

Results

5) Fig. 2: The power data shown in these graphs possibly indicate the averaged power data across the gait cycle (which possible calculated from the equation shown in Model definition section). If so, what is the conclusion obtained from this result? Please clarify.

6) Fig. 3: Which graph exhibit the results for ankle, knee, and hip power? Please indicate.

Discussion

7) The authors discussed the reason why the A2 peak had a peak at 60 yrs from the viewpoint of statistical technique. The authors should add some discussion on the biomechanical reason of the increased A2 peak at 60 yrs and the reason for no inverse-U shaped function across the age.

8) Taylor study only performed shod-walking and the other three studies were unshod-walking. The lower limb joint power may be different between unshod and shod walking. Thus, the authors should add this difference as limitation.

6. PLOS authors have the option to publish the peer review history of their article (what does this mean?). If published, this will include your full peer review and any attached files.

Reviewer #1: **Yes: **L. Eduardo Cofré Lizama

Reviewer #2: No

---

## [Author Response · Author response to Decision Letter 0]

3 Sep 2021

Please see the response in the attached document, where there is proper formatting to see the tables.

Reviewer #1

I think this is a great effort to elucidate the (linear/non-linear) relationship between ageing and gait mechanics "shift" as a continuum. The secondary data analysis approach is novel and worthy considering that several studies have already collected data to test similar hypotheses. Congratulations.

Reply: We thank the Reviewer for the positive comments.

However, although considered and mitigated (not mentioned to what extent) I have concerns in regards to the mixture of methodologies. There is 1 shod/treadmill study, 1 unshod/overground and 1 shod/overground; both shoes and treadmill have an effect on walking mechanics. I think a stronger argument to put these studies together is needed. To my knowledge, there is more evidence against than for putting studies with different methodologies together. 

Reply: We thank the Reviewer for this comment. When planning the design of the study, we considered whether data pooling would be appropriate given the reasons mentioned by the Reviewer. When pooling data across studies, be it in a meta-analysis or using individual data like our study, there is a need to balance the advantages of increasing sample size and the disadvantage of introducing excessive heterogeneity. Our decision was based on several factors:

1. We tried to emulate the methodology of a recent systematic review (Boyer et al., 2017), which pooled data into a random-effects model of studies which included barefoot (Cofre et al., 2011) and shod walking (Kulmala et al., 2014). When we reviewed Figure 4 of (Boyer et al., 2017) which was a pooled analysis of joint power data, we did not identify a consistent trend effect of shoes on joint power, resulting in our decision to pool the studies together.

2. In running, where most of the barefoot vs shod comparisons have been made, it has been reported that the primary effect of shoe type on running mechanics is due to the alteration in running pattern (Shih et al., 2013). Walking without shoes will reduce step length compared to shoes (Wirth et al., 2011). Whether the effect of footwear type on walking mechanics is attributed to alterations in the walking pattern is uncertain. Given that we also included stride length as a covariate, in addition to the random effects of the study, we felt that our analysis is justified.

3. In a study that directly compared treadmill vs overground walking, the authors reported no significant differences in A2, K2, and H3 (see Table 4 of (Lee and Hidler, 2008)), again providing us evidence that data pooling was appropriate.

Also I think that a table with full description of participants and methodologies, for comparison purposes, should be provided. What data set provided more information for the youngest and the oldest group? 

Reply: We thank this Reviewer and Reviewer 2 for asking us to put more details into the tables.

For Table 1, we provided the minimum and maximum ages for each study as suggested by Reviewer 2. We also included as supplementary material (Table SM1), a table of the data collection protocols for each study. 

Table 1. Descriptive characteristics (mean [standard deviation] for continuous variables) of participants 

Variables fukuchi (N = 33) horst (N = 57) schreiber (N = 48) taylor (N = 140)

Age (yo) 39.42 (17.87) 23.12 (2.73) 38.17 (13.97) 65.40 (6.47)

Age (min) 21 19 19 55

Age (Max) 84 30 67 86

Height (m) 1.67 (0.12) 1.74 (0.10) 1.74 (0.09) 1.68 (0.09)

Mass (kg) 67.66 (12.44) 67.93 (11.26) 71.96 (12.19) 74.03 (14.92)

Sex-F 15 (45%) 29 (51%) 23 (48%) 90 (64%)

Sex-M 18 (55%) 28 (49%) 25 (52%) 50 (36%)

Speed (m/s) 1.23 (0.17) 1.45 (0.10) 1.16 (0.14) 1.41 (0.19)

Stride length (m) 1.22 (0.14) 1.51 (0.06) 1.28 (0.12) 1.47 (0.16)

Table SM1. Brief methodologies of included studies

 Fukuchi et al. Horst et al. Schriber & Moissnenet Taylor et al.

Country Brazil Germany Luxembourg England

Inclusion criteria Healthy, free from lower limb injuries Physically active, without gait pathology and free of lower extremity injuries Asymptomatic, i.e. healthy and injury free for both lower and upper extremities Live independently, be independent walkers, with no surgical procedures

Camera system 12 cameras (Raptor-4, Motion Analysis Corporation, Santa Rosa, CA, USA) 10 cameras (Oqus 310, Qualisys, Gothenburg, Sweden) 10 cameras (Oqus 4, Qualisys, Gothenburg, Sweden) 7 cameras (T20, Vicon, Oxford, UK)

Camera sampling frequency 150Hz 250Hz 100Hz 100Hz

Force platform system Dual-belt, instrumented treadmill (FIT, Bertec, Columbus, OH, USA) 2 force plates (Type 9287CA, Kistler, Switzerland) 2 force plates (OR6-5, AMTI, Massachusetts, USA) 1 force plate (Type 9281CA, Kistler, Winterthur, Switzerland)

Force sampling frequency 300Hz 1000Hz 1500Hz 1000Hz

Footwear Unshod Unshod Unshod Shod

Surface Treadmill Overground Overground Overground

averaged power curves are quite different particularly at the hip (also the knee).

Reply: We thank the Reviewer for this comment. We agree with the Reviewers that variability at the hip and knee are high. During the data processing stage, we ensured that the signals were correct. Hence, we attribute this variability to the well-established issues such as soft-tissue artifact (hence, the variability visually reduces from the hip, knee, to ankle) and marker placement variability. 

I think that the authors hypothesis of a inverted U shape for joint power/work in relation to aging should be better explained considering that gait maturation occurs before (potentially the ascending part of the curve) the age range included.

Reply: We thank the Reviewer for this comment. Generating a “nonlinear” hypothesis is more challenging than a “linear” hypothesis. This is because the nonlinear relationship can take on many forms, from a simple U-shaped function to a function with multiple peaks and troughs. Given that there are no prior studies that have investigated the biomechanics of walking at smaller age intervals in adults, it is difficult to know when certain biomechanical peaks relative to age. 

The Reviewer is correct that some studies have shown that some joint kinetic variables stabilize by age 4 to 7 years old (Samson et al., 2013). However, a study reported that adults aged 23 to 31 years (Wu et al., 2019), walked with a dimensionless A2 joint power of 0.08 their at self-selected speed (we used http://markummitchell.github.io/engauge-digitizer/ to digitize Figure 4 of (Wu et al., 2019)). This is visually greater than the A2 value reported by 6-year-old children (Samson et al., 2013). In addition, some variables such as H3 did not plateau by age 6 years old (Samson et al., 2013). Based on the evidence, we find there is convincing prior knowledge to make a hypothesis that all joint powers peaked at the earliest age range investigated. 

Between the speeds investigated presently, joint kinetics typically scale with speed – joint kinetics increase in magnitudes at faster speeds (Zelik and Kuo, 2010). Given that walking speed appears to peak at 30 years old (Bohannon and Williams Andrews, 2011), we generated a global null hypothesis of an inverted U-shaped relationship between peak joint powers and age.

We modified the section in L85, which reads as:

Between the speeds investigated presently, joint kinetics typically scale with speed – joint kinetics increase in magnitudes at faster speeds [17]. Given that walking speed appears to peak at 30 years old [9], we generated the following null hypotheses: that A2 would exhibit an inverse “U” shaped function across age, with peaks at approximately 30 years old, similar to walking speed. Given that the hip and ankle power has a reciprocal relationship [4], we also hypothesized that H1 and H3 would exhibit a “U” shaped function across age, with a trough happening at the age where A2 peaks.

I partly disagree with the term "unnatural" (line 74) when asking subjects to walk at specific speeds considering that data collected in the lab is usually collected for short periods (5-8 strides unless on a treadmill), which may be similar to speed changes (range) in daily life when dealing with different street or home contexts.

Reply: We thank the Reviewer for this comment. We also agree that short periods of fixed speed walking may not necessarily result in a clinically significant alternation in walking pattern, relative to the same self-determined speed. A study reported that the biomechanics of walking was statistically, but not necessarily clinically, different between self-determined and fixed speed walking, at the same speed (Sloot et al., 2014). However, whether these differences would change at different walking speeds has not been investigated. 

In a systematic review of young adults walking, it was reported that artificial slow-walking resulted in lower cadence and shorter step length compared to comfortable speed walking (Fukuchi et al., 2019). However, even though older adults do walk slower than younger adults, the alterations may not be similar to experimental means of inducing aging. For example, one study reported that older adults walked with a shorter step length but with similar cadence than younger adults (Judge et al., 1996). 

In summary, we believe that whether fixed-speed walking is “unnatural” lies on a continuum and is dependent on the difference in magnitude relative to an individual’s self-determined speed. The bigger the difference, the more unnatural walking would become.

Reviewer #2

 This study investigated the effect of aging on the lower limb joint power using (non) linear statistical technique GAMLSS. The secondary analysis using GAMLSS method indicated that A2 peaks showed peak at 60yrs, which has not been supported by previous studies. I think the authors should discuss this result more from the biomechanical aspects. 

Reply: We thank the Reviewer for this comment. We have added another paragraph to explain our findings of A2. This can be found in L315:

Qiao and Jindrich proposed that joints/muscle groups could take on four different mechanical functional roles – spring, motor, damper, and strut [37]. It may be that a within-cycle adjusted A2 power has a different mechanical functional representation from the unadjusted raw A2. A joint’s total positive power could be derived from recycling energy from the elastic components of the muscle-tendon unit (i.e. joint as a spring), and/or purely from the concentric activity of a muscle (i.e. joint as a motor) [37]. Speculatively, the raw A2 variable may represent the ankle’s total power. Also, the within-cycle adjusted A2 may present power derived from the motor-function of the joint, since what is left after adjusting away for negative power, is the main effect of peak positive power. As previously mentioned, given the decline in Achilles tendon stiffness with age [36], our results could be interpreted as an augmentation of ankle motor-function with age for propulsion. If aging results in an elevation of ankle motor-function, at the expense of spring-function, this could explain the decline in mechanical efficiency in walking with increasing age [38].

Other comments are as follows.

Introduction

１）What was the rationale behind hypothesizing inverse U shaped function for A2 peak and U shaped function for H1 and H3 peaks across age? Supporting literature is needed for these hypotheses.

Reply: We thank the Reviewer for this comment, which is identical to a comment by Reviewer 1. We have addressed this in detail in the response to Reviewer 1, and have modified the section in L85, which reads as:

Between the speeds investigated presently, joint kinetics typically scale with speed – joint kinetics increase in magnitudes at faster speeds [17]. Given that walking speed appears to peak at 30 years old [9], we generated the following null hypotheses: that A2 would exhibit an inverse “U” shaped function across age, with peaks at approximately 30 years old, similar to walking speed. Given that the hip and ankle power has a reciprocal relationship [4], we also hypothesized that H1 and H3 would exhibit a “U” shaped function across age, with a trough happening at the age where A2 peaks.

Methods

2) Table 1: Please add the range of the age of participants in each study.

Reply: We have added the age range toof the participants in each study Table 1.

Table 1. Descriptive characteristics (mean [standard deviation] for continuous variables) of participants 

Variables fukuchi (N = 33) horst (N = 57) schreiber (N = 48) taylor (N = 140)

Age (yo) 39.42 (17.87) 23.12 (2.73) 38.17 (13.97) 65.40 (6.47)

Age (min) 21 19 19 55

Age (Max) 84 30 67 86

Height (m) 1.67 (0.12) 1.74 (0.10) 1.74 (0.09) 1.68 (0.09)

Mass (kg) 67.66 (12.44) 67.93 (11.26) 71.96 (12.19) 74.03 (14.92)

Sex-F 15 (45%) 29 (51%) 23 (48%) 90 (64%)

Sex-M 18 (55%) 28 (49%) 25 (52%) 50 (36%)

Speed (m/s) 1.23 (0.17) 1.45 (0.10) 1.16 (0.14) 1.41 (0.19)

Stride length (m) 1.22 (0.14) 1.51 (0.06) 1.28 (0.12) 1.47 (0.16)

3) Common processing across four studies: Each participant would conducted multiple walking trials. Was the joint power data averaged across the walking trials?

Reply: Yes, the joint power data was averaged across multiple strides of each person at a particular walking speed. We have added the sentence below:

L165

For each participant and speed, the average joint power across multiple strides was calculated.

4) Model definition: what does “j” mean?

Reply: We have reword the meaning of j, as seen below:

where yij is the power value of the respective joint of the ith subject and jth gait cycle point

Results

5) Fig. 2: The power data shown in these graphs possibly indicate the averaged power data across the gait cycle (which possible calculated from the equation shown in Model definition section). If so, what is the conclusion obtained from this result? Please clarify.

Reply: We thank the Reviewer for this comment. We have provided a better explanation of Figure 2 in:

L259

Figure 2 depicts the modelled smooth effect of age against joint power, which can be interpreted as the main effect of age on the average power marginalized across the gait cycle. The clearest trend with age was the knee, which saw a shift from an average positive power at 19 years old, to an average negative value peaking at -0.09 (95%CI -0.12 to -0.06) which occurred at 68 years of age, followed by a shift back to an average positive power thereafter. The smooth effect of age on power had no clear trends for the ankle and hips joints, where the 95%CI included the zero value across the age spectrum investigated.

We also provided a physiological interpretation of the results in L328:

The knee has not been thought of as an active energy source for propulsion in walking [39], but is important for shock absorption, joint stability, and inter-segmental energy transfer. The shift in knee power from an average positive from 19 years old to an average negative value peaking at 68 years old (Figure 2), potentially reflects an age-related biasing of muscle absorption over muscle generation. Greater negative than positive work with aging could suggest that the knee is behaving more like a damper with age [37], with the ensuring result that more positive work has to be performed by adjacent muscles to maintain walking speed. After 70 years old, the shift in knee average power from negative to positive coincides with the decline in A2 power (Figure 4), suggesting that the knee may be compensating for age-related propulsive deficits from the ankle. Evidently, research into how age influences the different mechanical functions within the lower-limb joints may provide a better understanding of what impairments drive a decline in walking performance.

6) Fig. 3: Which graph exhibit the results for ankle, knee, and hip power? Please indicate.

Reply: We apologize for missing out on this information. We have included this in the revised caption for Figure 3 in L280.

Predicted mean joint power waveform from the GAMLSS model at two walking speeds (1 and 1.5m/s), at each of the seven age groups, at a fixed stride length of 1.5m. (a) Ankle power, (b) Knee power, (c) Hip power.

Discussion

7) The authors discussed the reason why the A2 peak had a peak at 60 yrs from the viewpoint of statistical technique. The authors should add some discussion on the biomechanical reason of the increased A2 peak at 60 yrs and the reason for no inverse-U shaped function across the age.

Reply: We have addressed this in response to an earlier comment by the Reviewer. The added discussion can be found in L315 of the Discussion.

Qiao and Jindrich proposed that joints/muscle groups could take on four different mechanical functional roles – spring, motor, damper, and strut [37]. It may be that a within-cycle adjusted A2 power has a different mechanical functional representation from the unadjusted raw A2. A joint’s total positive power could be derived from recycling energy from the elastic components of the muscle-tendon unit (i.e. joint as a spring), and/or purely from the concentric activity of a muscle (i.e. joint as a motor) [37]. Speculatively, the raw A2 variable may represent the ankle’s total power. Also, the within-cycle adjusted A2 may present power derived from the motor-function of the joint, since what is left after adjusting away for negative power, is the main effect of peak positive power. As previously mentioned, given the decline in Achilles tendon stiffness with age [36], our results could be interpreted as an augmentation of ankle motor-function with age for propulsion. If aging results in an elevation of ankle motor-function, at the expense of spring-function, this could explain the decline in mechanical efficiency in walking with increasing age [38].

8) Taylor study only performed shod-walking and the other three studies were unshod-walking. The lower limb joint power may be different between unshod and shod walking. Thus, the authors should add this difference as limitation.

Reply: We have added this as a limitation, which reads as in L387:

A second limitation of the present analysis was the inclusion of studies with heterogeneous experimental set-ups, such as overground [21] versus treadmill walking [19], shod (taylor data) versus unshod conditions [21]. However, between-study heterogeneity in experimentation and indeed between-subject heterogeneity is a common occurrence in secondary analysis studies, such as in a meta-analysis [3]. We mitigated this issue by including study-specific and subject-specific random effects into our models. This meant that our predictive estimates were marginalized over different experimental study protocols and participants.

We would also like to direct the Reviewer to the response to Reviewer 1 on the issue of pooling data across studies, which also touches on this issue of footwear conditions.

References

Bohannon, R.W., Williams Andrews, A., 2011. Normal walking speed: a descriptive meta-analysis. Physiotherapy 97, 182-9.

Boyer, K.A., Johnson, R.T., Banks, J.J., Jewell, C., Hafer, J.F., 2017. Systematic review and meta-analysis of gait mechanics in young and older adults. Exp Gerontol 95, 63-70.

Cofre, L.E., Lythgo, N., Morgan, D., Galea, M.P., 2011. Aging modifies joint power and work when gait speeds are matched. Gait Posture 33, 484-9.

Fukuchi, C.A., Fukuchi, R.K., Duarte, M., 2019. Effects of walking speed on gait biomechanics in healthy participants: a systematic review and meta-analysis. Syst Rev 8, 153.

Judge, J.O., Õunpuu, S., Davis, R.B., 1996. Effects of Age on the Biomechanics and Physiology of Gait. Clin Geriatr Med 12, 659-78.

Kulmala, J.P., Korhonen, M.T., Kuitunen, S., Suominen, H., Heinonen, A., Mikkola, A., Avela, J., 2014. Which muscles compromise human locomotor performance with age? J R Soc Interface 11, 20140858.

Lee, S.J., Hidler, J., 2008. Biomechanics of overground vs. treadmill walking in healthy individuals. J Appl Physiol (1985) 104, 747-55.

Samson, W., Van Hamme, A., Desroches, G., Dohin, B., Dumas, R., Chèze, L., 2013. Biomechanical maturation of joint dynamics during early childhood: updated conclusions. J Biomech 46, 2258-63.

Shih, Y., Lin, K.-L., Shiang, T.-Y., 2013. Is the foot striking pattern more important than barefoot or shod conditions in running? Gait Posture 38, 490-4.

Sloot, L.H., van der Krogt, M.M., Harlaar, J., 2014. Self-paced versus fixed speed treadmill walking. Gait Posture 39, 478-84.

Wirth, B., Hauser, F., Mueller, R., 2011. Back and neck muscle activity in healthy adults during barefoot walking and walking in conventional and flexible shoes. Footwear Science 3, 159-67.

Wu, A.R., Simpson, C.S., van Asseldonk, E.H.F., van der Kooij, H., Ijspeert, A.J., 2019. Mechanics of very slow human walking. Sci Rep 9, 18079.

Zelik, K.E., Kuo, A.D., 2010. Human walking isn't all hard work: evidence of soft tissue contributions to energy dissipation and return. J Exp Biol 213, 4257-64.

---

## [Decision Letter · Decision Letter 1]

13 Oct 2021

PONE-D-21-18361R1The mechanical energetics of walking across the adult lifespan.PLOS ONE

Dear Dr. Liew,

Thank you for submitting your manuscript to PLOS ONE. After careful consideration, we feel that it has merit but does not fully meet PLOS ONE’s publication criteria as it currently stands. Therefore, we invite you to submit a revised version of the manuscript that addresses the points raised during the review process.

ACADEMIC EDITOR: Please follow the reviewer 2's suggestions on the addition to the introduction/methods. 

We look forward to receiving your revised manuscript.

Kind regards,

Kei Masani

Academic Editor

PLOS ONE

Journal Requirements:

Additional Editor Comments (if provided):

Reviewers' comments:

Reviewer's Responses to Questions

**Comments to the Author**

1. If the authors have adequately addressed your comments raised in a previous round of review and you feel that this manuscript is now acceptable for publication, you may indicate that here to bypass the “Comments to the Author” section, enter your conflict of interest statement in the “Confidential to Editor” section, and submit your "Accept" recommendation.

Reviewer #1: All comments have been addressed

Reviewer #2: All comments have been addressed

2. Is the manuscript technically sound, and do the data support the conclusions?

Reviewer #1: Yes

Reviewer #2: Yes

3. Has the statistical analysis been performed appropriately and rigorously? 

Reviewer #1: Yes

Reviewer #2: Yes

4. Have the authors made all data underlying the findings in their manuscript fully available?

Reviewer #1: Yes

Reviewer #2: Yes

5. Is the manuscript presented in an intelligible fashion and written in standard English?

Reviewer #1: Yes

Reviewer #2: Yes

6. Review Comments to the Author

Reviewer #1: First of al, my apologies for my delayed response. Thanks to the authors for replying to my comments regarding the manuscript. I think that some of the replies, particularly those related to the differences (tread vs overground and shod and unshod) between studies and the rationale for putting them together should be part of the introduction or at least should be address in the methods (not only in the limitations). I really appreciate authors present their findings in relation to the contrast with most literature in the field. This study will be certainly controversial, even more considering potential implications for rehabilitation.

Minor

I think you should put studies by surname with capital letter?

Line 188: typo. “gaitcycle”

Reviewer #2: (No Response)

7. PLOS authors have the option to publish the peer review history of their article (what does this mean?). If published, this will include your full peer review and any attached files.

Reviewer #1: **Yes: **L. Eduardo Cofré Lizama

Reviewer #2: No

---

## [Author Response · Author response to Decision Letter 1]

13 Oct 2021

Please see Response in the pdf proof as it has the appropriate formatting and images included.

Journal Requirements

Reply: We are unaware that any of our citations are that of retracted articles. We kindly request the Editorial team to flag such an instance if an error has been made.

Reviewer #1

Reviewer #1: First of al, my apologies for my delayed response. Thanks to the authors for replying to my comments regarding the manuscript. I think that some of the replies, particularly those related to the differences (tread vs overground and shod and unshod) between studies and the rationale for putting them together should be part of the introduction or at least should be address in the methods (not only in the limitations). I really appreciate authors present their findings in relation to the contrast with most literature in the field. This study will be certainly controversial, even more considering potential implications for rehabilitation.

Reply: We thank the Reviewer for this comment. We agree that this is a controversial paper, hence we aim to be very transparent in our analysis. We have included the rationale in the rationale in the Methods, which reads as in L97:

Despite the presence of some methodological variations between the presently included studies, data pooling was deemed appropriate to conduct based on several reasons. First, a previous meta-analysis [3] pooled data into a random-effects model despite methodological variations in the primary studies (e.g. barefoot walking [6] and shod walking [22]). The present analysis also adopted a random effects modelling approach. Second, a previous study reported no significant differences in A2 and H3 powers between treadmill and overground walking [23]. Third, given that walking without shoes reduces step length compared to shoes [24], to account for between-study variation in footwear presently, we included step length as a covariate in our models. 

We have also included in L76 of the Introduction a sentence on our study design:

To achieve this aim, we pooled together the individual participant data of three publicly available datasets [16-18], and the data from one primary research.

Minor

I think you should put studies by surname with capital letter?

Reply: We thank the Reviewer for this comment. We apologise to the Reviewer because we are uncertain as to the source of the problem. Is the Reviewer referring to any in-text citations, or the formatting of the bibliography? We have tried to follow all formatting to Plos ONE requirements.

Example of our formatting below

Example of a Plos ONE publication (doi: 10.1371/journal.pone.0238690)

Line 188: typo. “gaitcycle”

Reply: We have corrected this to read as “gait cycle”.

---

## [Editor Report · Decision Letter 2]

27 Oct 2021

The mechanical energetics of walking across the adult lifespan.

PONE-D-21-18361R2

Dear Dr. Liew,

We’re pleased to inform you that your manuscript has been judged scientifically suitable for publication and will be formally accepted for publication once it meets all outstanding technical requirements.

Kind regards,

Kei Masani

Academic Editor

PLOS ONE
---

## [Editor Report · Acceptance letter]

3 Nov 2021

PONE-D-21-18361R2 

The mechanical energetics of walking across the adult lifespan. 

Dear Dr. Liew:

I'm pleased to inform you that your manuscript has been deemed suitable for publication in PLOS ONE. Congratulations! Your manuscript is now with our production department. 

Kind regards, 

on behalf of

Dr. Kei Masani 

Academic Editor

PLOS ONE